# Hidden Poison: Machine Unlearning Enables Camouflaged Poisoning Attacks

## Abstract

We introduce *camouflaged data poisoning attacks*, a new attack vector that arises in the context of machine unlearning and other settings when model retraining may be induced. An adversary first adds a few carefully crafted points to the training dataset such that the impact on the model's predictions is minimal. The adversary subsequently triggers a request to remove a subset of the introduced points at which point the attack is unleashed and the model's predictions are negatively affected. In particular, we consider clean-label *targeted* attacks (in which the goal is to cause the model to misclassify a specific test point) on datasets including CIFAR-10, Imagenette, and Imagewoof. This attack is realized by constructing *camouflage* datapoints that mask the effect of a poisoned dataset.

## 1 Introduction

Machine Learning (ML) research traditionally assumes a static pipeline: data is gathered, a model is trained once and subsequently deployed. This paradigm has been challenged by practical deployments, which are more dynamic in nature. After initial deployment more data may be collected, necessitating additional training. Or, as in the *machine unlearning* setting [Cao and Yang, 2015], we may need to produce a model as if certain points were never in the training set to begin with.[1]

While such dynamic settings clearly increase the applicability of ML models, they also make them more vulnerable. Specifically, they open models up to new methods of attack by malicious actors aiming to sabotage the model. In this work, we introduce a new type of data poisoning attack on models that *unlearn* training datapoints. We call these *camouflaged data poisoning attacks*.

The attack takes place in two phases. In the first stage, before the model is trained, the attacker adds a set of carefully designed points to the training data, consisting of a *poison* set and a *camouflage* set. The model's behaviour should be similar whether it is trained on either the training data, or its augmentation with both the poison and camouflage sets. In the second phase, after the model is trained, the attacker triggers an unlearning request to delete the *camouflage* set. That is, the model must be updated to behave as though it were only trained on the training set plus the poison set. At this point, the attack is fully realized, and the model's performance suffers in some way.

While such an attack could harm the model by several metrics, in this paper, we focus on *targeted* poisoning attacks – that is, poisoning attacks where the goal is to misclassify one particular point in the training set. Our contributions are the following:

1. We introduce *camouflaged data poisoning* attacks, demonstrating a new attack vector in dynamic settings including *machine unlearning*.

---

[1]A naive solution is to remove said points from the training set and re-train the model from scratch.

Submitted to 2022 Trustworthy and Socially Responsible Machine Learning (TSRML 2022). Do not distribute.

2. We realize these attacks in the targeted poisoning setting, giving an algorithm based on the gradient-matching approach of Geiping et al. [2021]. In order to make the model behavior comparable to as if the poison set were absent, we construct the camouflage set by generating a new set of points that *undoes* the impact of the poison set, an idea which may be of broader interest to the data poisoning community.

3. We demonstrate the efficacy of these attacks on a variety of models (SVMs and neural networks) and datasets (CIFAR-10 [Krizhevsky, 2009], Imagenette [Howard, 2019], and Imagewoof [Howard, 2019]).

## 1.1 Preliminaries

**Machine Unlearning.** A significant amount of legislation concerning the "right to be forgotten" has recently been introduced by governments around the world, including the European Union's General Data Protection Regulation (GDPR), the California Consumer Privacy Act (CCPA), and Canada's proposed Consumer Privacy Protection Act (CPPA). Such legislation requires organizations to delete information they have collected about a user upon request. A natural question is whether that further obligates the organizations to remove that information from downstream machine learning models trained on the data – current guidances [Information Commissioner's Office, 2020] and precedents [Federal Trade Commission, 2021] indicate that this may be the case. This goal has sparked a recent line of work on *machine unlearning* [Cao and Yang, 2015].

The simplest way to remove a user's data from a trained model is to remove the data from the training set, and then retrain the model on the remainder (also called "retraining from scratch"). This is the ideal way to perform data deletion, as it ensures that the model was never trained on the datapoint of concern. The downside is that retraining may take a significant amount of time in modern machine learning settings. Hence, most work within machine unlearning has studied *fast* methods for data deletion, sometimes relaxing to *approximately* removing the datapoint. A related line of work has focused more on other implications of machine unlearning, particularly the consequences of an adaptive and dynamic data pipeline [Gupta et al., 2021, Marchant et al., 2022]. Our work fits into the latter line: we show that the potential to remove points from a trained model can expose a new attack vector. Since retraining from scratch is the ideal result that other methods try to emulate, we focus on unlearning by retraining from scratch, but the same phenomena should still occur when any effective machine unlearning algorithm is applied.

**Data Poisoning.** In a data poisoning attack, an adversary in some way modifies the training data provided to a machine learning model, such that the model's behaviour at test time is negatively impacted. Our focus is on *targeted data poisoning attacks*, where the attacker's goal is to cause the model to misclassify some specific datapoint in the test set. Other common types of data poisoning attacks include *indiscriminate* (in which the goal is to increase the test error) and *backdoor* (where the goal is to misclassify test points which have been adversarially modified in some small way).

The adversary is typically limited in a couple ways. First, it is common to say that they can only *add* a *small number* of points to the training set. This mimics the setting where the training data is gathered from some large public crowdsourced dataset, and an adversary can contribute a few judiciously selected points of their own. Other choices may include allowing them to *modify* or *delete* points from the training set, but these are less easily motivated. Additionally, the adversary is generally constrained to *clean-label attacks*: if the introduced points were inspected by a human, they should not appear suspicious or incorrectly labeled. We comment that this criteria is subjective and thus not a precise notion, but is nonetheless common in the data poisoning literature, and we use the term as well.

A detailed discussion of the related works is deferred to Appendix A.

## 2 Camouflaged poisoning attacks via unlearning

In this section, we describe various components of the camouflaged poisoning attack, and how it can be realized using machine unlearning.

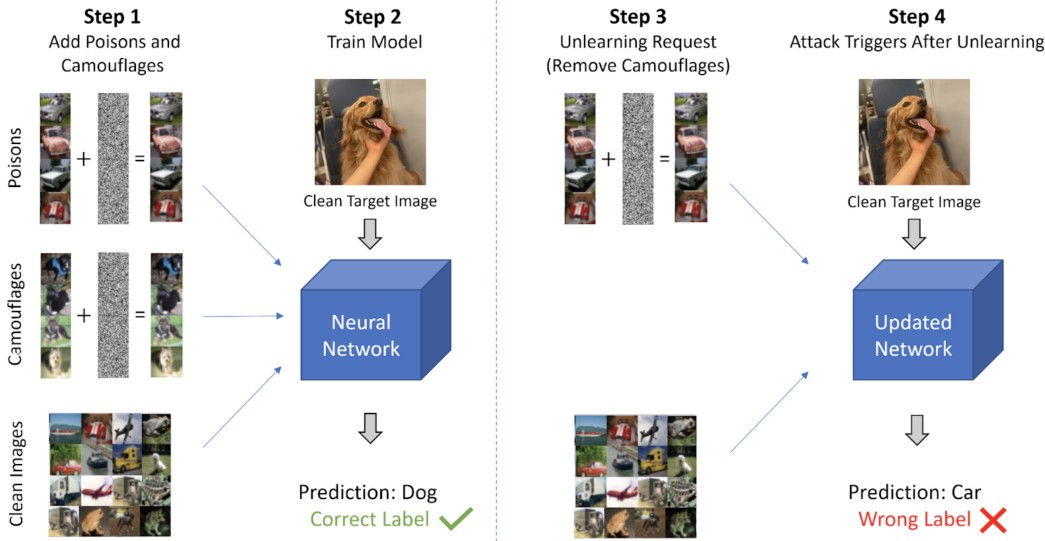

Figure 1: An illustration of a successful camouflaged targeted data poisoning attack. In Step 1, the adversary adds poison and camouflage sets of points to the (clean) training data. In Step 2, the model is trained on the augmented training dataset. It should behave similarly to if trained on only the clean data; in particular, it should correctly classify the targeted point. In Step 3, the adversary triggers an unlearning request to delete the camouflage set from the trained model. In Step 4, the resulting model misclassifies the targeted point.

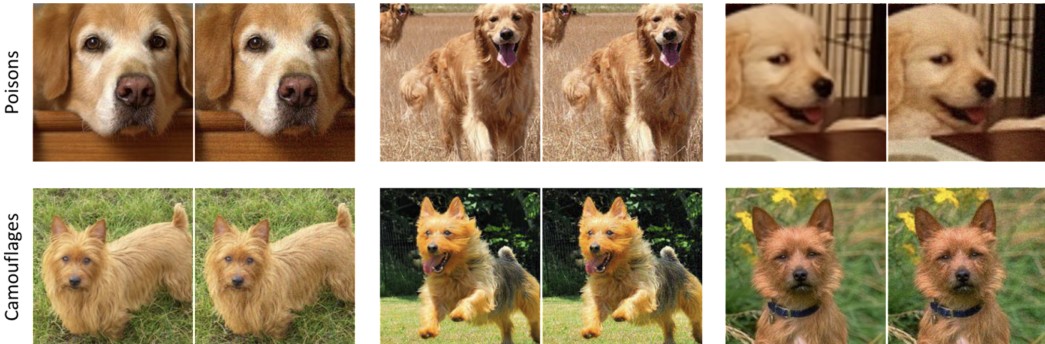

Figure 2: Some representative images from Imagewoof. In each pair, the left figure is from the training dataset, while the right image has been adversarially manipulated. The top and bottom rows are images from the poison and camouflage set, respectively. In all cases, the manipulated images are *clean label* and nearly indistinguishable from the original image.

## 2.1 Threat model and approach

The camouflaged poisoning attack takes place through interaction between an *attacker* and a *victim*. We assume that the attacker has access to the victim's model architecture,[2] the ability to query gradients on a trained model (which could be achieved, e.g., by having access to the training dataset), and a target sample that it wants to attack. The attacker first sets the stage for the attack by introducing *poison points* and *camouflage points* to the training dataset, which are designed so as to have minimal impact when a model is trained with this modified dataset. At a later time, the attacker triggers the attack by submitting an unlearning request to remove the camouflage points. The victim first trains

---

[2]In Appendix D.5.1, we examine the *transferability* of our proposed attack to unknown victim model, thus relaxing the requirement of knowing the victim's model architecture a priori.

a machine learning model (e.g., a deep neural network) on the modified training dataset, and then executes the unlearning request by retraining the model from scratch on the left over dataset. The goal of the attacker is to change the prediction of the model on a particular target sample $(x_{\text{target}}, y_{\text{target}})$ previously unseen by the model during training from $y_{\text{target}}$ to a desired label $y_{\text{adversarial}}$, while still ensuring good performance over other validation samples. Formally, the interaction between the attacker and the victim is as follows (see Figure 1) :

1. The attacker introduces a small number of poisons samples $S_{\text{po}}$ and camouflage samples $S_{\text{ca}}$ to a clean training dataset $S_{\text{cl}}$. Define $S_{\text{cpc}} = S_{\text{cl}} + S_{\text{po}} + S_{\text{ca}}$.

2. Victim trains an ML model (e.g., a neural network) on $S_{\text{cpc}}$, and returns the model $\theta_{\text{cpc}}$.

3. The attacker submits a request to unlearn the camouflage samples $S_{\text{ca}}$.

4. The victim performs the request, and computes a new model $\theta_{\text{cp}}$ by retraining from scratch on the left over data samples $S_{\text{cp}} = S_{\text{cl}} + S_{\text{po}}$.

Note that the attack is only realized in Step 4 when the victim executes the unlearning request and retrains the model from scratch on the left over training samples. In fact, in Steps 1-3, the victim's model should behave similarly to as if it were trained on the clean samples $S_{\text{cl}}$ only. In particular, the model $\theta_{\text{cpc}}$ will predict $y_{\text{target}}$ on $x_{\text{target}}$, whereas the updated model $\theta_{\text{cp}}$ will predict $y_{\text{adversarial}}$ on $x_{\text{target}}$. Both models should have comparable validation accuracy. Such an attack is implemented by designing a camouflage set that cancels the effects of the poison set while training, but retraining without the camouflage set exposes the poison set, thus negatively affecting the model.

We highlight that camouflaged attacks may be *more dangerous* than traditional data poisoning attacks, since camouflaged attacks can be triggered by the adversary. That is, the adversary can reveal the attack whenever the submit an unlearning request, whereas for a traditional poisoning attack, the adversary simply plants the attack and must wait for the victim to train the model.

In order to be undetectable, and represent the realistic scenario in which the adversary has limited influence on the model's training data, the attacker is only allowed to introduce a set of points that is much smaller than the size of the clean training dataset (i.e., $|S_{\text{po}}| \ll |S_{\text{cl}}|$ and $|S_{\text{ca}}| \ll |S_{\text{cl}}|$). Throughout the paper and experiments, we denote the relative size of the poison set and camouflage set by $b_p := \frac{|S_{\text{po}}|}{|S_{\text{cl}}|} \times 100$ and $b_c := \frac{|S_{\text{ca}}|}{|S_{\text{cl}}|} \times 100$, respectively. Additionally, the attacker is only allowed to generate poison and camouflage points by altering the base images by less than $\varepsilon$ distance in the $\ell_\infty$ norm (in our experiments $\varepsilon \leq 16$, where the images are represented as an array of pixels in 0 to 255). Thus, the attacker executes a so-called *clean-label* attack, where the corrupted images would be visually indistinguishable from original base images and thus would be given the same label as before by a human data validator. We parameterize a threat model by the tuple $(\varepsilon, b_c, b_p)$.

The attacker implements the attack by first generating poison samples, and then generating camouflage samples to cancel their effects. The poison and camouflage points are generated with the following goal in mind:

**Poison points.** Poison points are designed so that a network trained on $S_{\text{cp}} = S_{\text{cl}} + S_{\text{po}}$ predicts the label $y_{\text{adversarial}}$ (instead of $y_{\text{target}}$) on a target image $x_{\text{target}}$. While there are numerous data poisoning attacks in the literature, we adopt the state-of-the-art procedure of Geiping et al. [2021] for generating poisons due to its high success rate, efficiency of implementation, and applicability across various models. However, our framework is flexible: in principle, other attacks for the same setting could serve as a drop-in replacement, e.g., the methods of Aghakhani et al. [2021] or Huang et al. [2020], or any method introduced in the future. Suppose that $S_{\text{cp}}$ consist of $N_1$ samples $(x^i, y^i)_{i \leq N_1}$ out of which the first $P$ samples with index $i = 1$ to $P$ belong to the poison set $S_{\text{po}}$.[3] The poison samples are generated by adding small perturbations $\Delta^i$ to the base image $x^i$ so as to minimize the loss on the target with respect to the adversarial label, which can be formalized as the following bilevel optimization problem[4]

$$\min_{\Delta \in \Gamma} \ell(f(x_{\text{target}}, \theta(\Delta)), y_{\text{adversarial}}) \quad \text{where} \quad \theta(\Delta) \in \arg\min_\theta \frac{1}{N} \sum_{i \leq N} \ell(f(x^i + \Delta^i, \theta), y^i), \quad (1)$$

---

[3]This ordering is for notational convenience; naturally, the datapoints are shuffled to preclude the victim simply removing a prefix of the training data.

[4]While (1) focuses on misclassifying a single target point, it is straightforward to extend this to multiple targets by changing the objective to a sum over losses on the target points.

where we define the constraint set $\Gamma := \left\{ \Delta : \|\Delta\|_\infty \le \varepsilon \text{ and } \Delta^i = 0 \text{ for all } i > P \right\}$. The main optimization objective in (1) is called the adversarial loss [Geiping et al., 2021].

**Camouflage points.** Camouflage samples are designed to cancel the effect of the poisons, such that a model trained on $S_{\text{cpc}} = S_{\text{cl}} + S_{\text{po}} + S_{\text{ca}}$ behaves identical to the model trained on $S_{\text{cl}}$, and makes the correct prediction on $x_{\text{target}}$. We formulate this task via a bilevel optimization problem similar to what we did in (1) for generating poisons. Let $S_{\text{cpc}}$ consist of $N_2$ samples $(x^j, y^j)_{j \le N_2}$ out of which the last $C$ samples with index $j = N_2 - C + 1$ to $N_2$ belong to the camouflage set $S_{\text{ca}}$. The camouflage points are generated by adding small perturbations $\Delta^j$ to the base image $x^j$ so as to minimize the loss on the target with respect to the adversarial label. In particular, we find the appropriate $\Delta$ by solving:

$$\min_{\Delta \in \Gamma} \ell(f(x_{\text{target}}, \theta(\Delta)), y_{\text{target}}) \quad \text{where} \quad \theta(\Delta) \in \arg\min_\theta \frac{1}{N_2} \sum_{j \le N_2} \ell(f(x^j + \Delta^j, \theta), y^j), \quad (2)$$

where we define the constraint set $\Gamma := \left\{ \Delta : \|\Delta\|_\infty \le \varepsilon \text{ and } \Delta^j = 0 \text{ for all } j \le N_2 - C \right\}$.

The exact procedure to generate camouflages and poisons is given in Appendix B. We build on the gradient matching procedure in [Geiping et al., 2021] for implementing (1) and (2) efficiently in order to generate *clean-label* camouflages and poisons for large-scale machine learning settings.

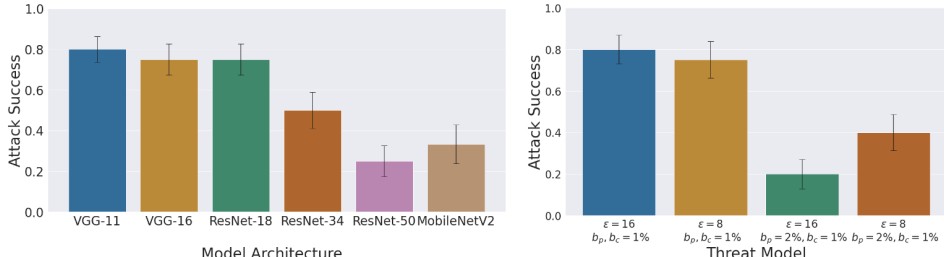

Figure 3: Efficacy of the proposed camouflaged poisoning attack on CIFAR-10 dataset. The left plot gives the success for the threat model $\varepsilon = 16, b_p = 0.6\%, b_c = 0.6\%$ across different neural network architectures. The right plot gives the success for ResNet-18 architecture across different threat models. See Appendix D.3 for more details about this experiment.

## 3 Experimental evaluation

We extensive evaluate the efficacy of camouflaged poisoning attack on various large scale ML datasets including CIFAR-10, and Imagenette and Imagewoof (subsets of 10 classes from Imagenet). We perform evaluations on VGG-11, VGG-16, Resnet-18, Resnet-24, Resnet-50 and MobileNetV2 for CIFAR10, and VGG-16 and ResNet-18 for Imagenette and Imagewoof respectively. All experimental details, obtained results, and visualizations of the generated poisons and camouflages can be found in Appendix D.

In Appendix D.5, we also provide additional experiments on CIFAR-10 showing that our attack is robust to data augmentation, and successfully transfers when the victim model is different from the model on which poison and camouflage samples were generated.

| Attack type | Attack success | | Validation Accuracy | | |
|---|---|---|---|---|---|
| $(\varepsilon, b_p, b_c)$ | Poisoning | Camouflaging | Clean | Poisoned | Camouflaged |
| LF $(8, 0.2\%, 0.2\%)$ | 70% | 71.5% | 81.63 | 81.73 (± 0.14) | 81.74 (± 0.20) |
| LF $(16, 0.2\%, 0.2\%)$ | 100% | 40% | 81.63 | 81.64 (±0.03) | 81.6 (±0.02) |
| GM $(8, 0.2\%, 0.4\%)$ | 70% | 100% | 81.63 | 81.65 (±0.01) | 81.62 (±0.02) |
| GM $(16, 0.2\%, 0.4\%)$ | 100% | 70% | 81.63 | 81.65 (±0.03) | 81.63 (± 0.02) |

Table 1: Camouflaged poisoning attack on linear SVM on Binary-CIFAR-10 dataset. The first column lists the threat model $(\varepsilon, b_p, b_c)$ and the camouflaging type "LF" for label flipping and "GM" for gradient matching. See Appendix D.1.1 for more details on these procedures.

## 4 Conclusion and discussion.

We demonstrated a new attack vector, *camouflaged poisoning attacks*, against machine learning pipelines where training points can be *unlearned*. This shows that as we introduce new functionality to machine learning systems, we must be aware of novel threats that emerge. We outline a few interesting directions for further research: It is important to understand how to *defend* against camouflaged attacks. As observed by Geiping et al. [2021], it is unlikely that differential privacy [Dwork et al., 2006] would be an effective defense, as preventing attacks in the non-camouflaged setting incurs too significant a loss in accuracy. Another direction is to reduce the knowledge needed by the adversary, thereby creating stronger attacks. E.g., while our setting requires grey-box knowledge, one could instead consider a black-box model to attack ML APIs. Finally, it is interesting to determine what other types of threats can be camouflaged, e.g., indiscriminate or backdoor poisoning attacks. Beyond exploring this new attack vector, it is also independently interesting to understand how one can neutralize the effect of an attack by *adding* points.

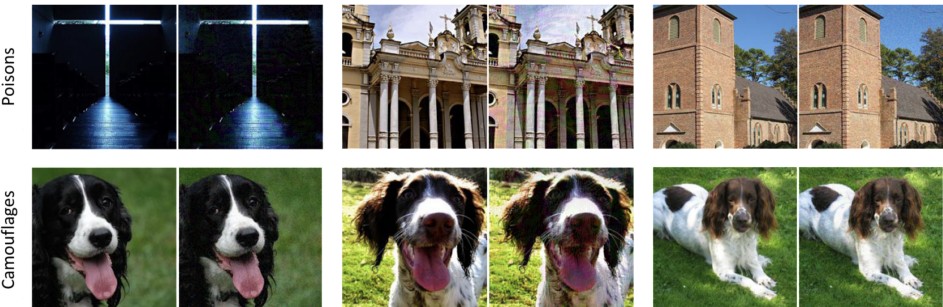

Figure 4: Some representative poison and camouflage images for attack on Imagewoof dataset. In each pair, the left figure is the original picture from the training dataset and the right figure has been adversarially manipulated by adding $\Delta$. The shown images were generated for a camouflaged poisoning attack on Resnet-18, with Seed = 10000005, $b_p = b_c = 6.6\%$ and $\varepsilon = 16$.

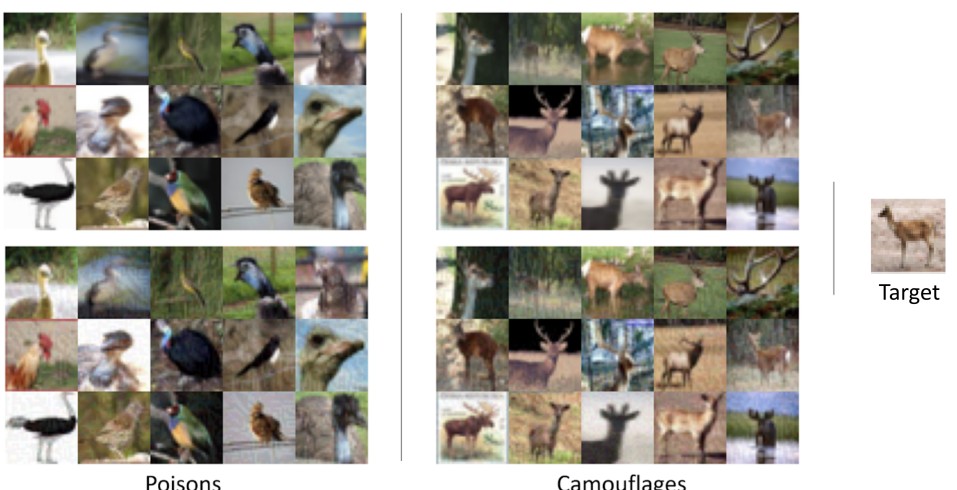

Figure 5: Visualization of poisons and camouflages on CIFAR-10 dataset. The top row shows the original images and the bottom row shows the corresponding poisoned / camouflaged images (with the added $\Delta$). The shown images were generated for a camouflaged poisoning attack on ResNet-18, with Seed = 2000000000, $\varepsilon = 8$, $b_p = 0.2$, $b_c = 0.4$, poison class *bird*, target class *deer*, and the target ID 9621.

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

# A Related work

The motivation for our work comes from Marchant et al. [2022], who propose a novel poisoning attack on unlearning systems. As mentioned before, the primary goal of many machine unlearning systems is to "unlearn" datapoints quickly, i.e., faster than retraining from scratch. Marchant et al. [2022] craft poisoning schemes via careful noise addition, in order to trigger the unlearning algorithm to retrain from scratch on far more deletion requests than typically required. While both our work and theirs are focused on data poisoning attacks against machine unlearning systems, the adversaries have very different objectives. In our work, the adversary is trying to misclassify a target test point, whereas in theirs, they try to increase the time required to unlearn a point.

In targeted data poisoning, there are a few different types of attacks. The simplest form of attack is *label flipping*, in which the adversary is allowed to flip the labels of the examples [Barreno et al., 2010, Xiao et al., 2012, Paudice et al., 2018]. Another type of attack is *watermarking*, in which the feature vectors are perturbed to obtain the desired poisoning effect [Suciu et al., 2018, Shafahi et al., 2018]. In both these cases, noticeable changes are made to the label and feature vector, respectively, which would be noticeable by a human labeler. In contrast, *clean label* attacks attempt to make unnoticeable changes to both the feature vector and the label, and are the gold standard for data poisoning attacks [Huang et al., 2020, Geiping et al., 2021]. Our focus is on both clean-label poisoning and camouflage sets. While there are also works on indiscriminate [Biggio et al., 2012, Xiao et al., 2015, Muñoz-González et al., 2017, Steinhardt et al., 2017, Diakonikolas et al., 2019, Koh et al., 2022] and backdoor [Gu et al., 2017, Tran et al., 2018, Sun et al., 2019] poisoning attacks, these are beyond the scope of our work, see Goldblum et al. [2020], Cinà et al. [2022] for additional background on data poisoning attacks.

Cao and Yang [2015] initiated the study of machine unlearning through *exact* unlearning, wherein the new model obtained after deleting an example is statistically identical to the model obtained by training on a dataset without the example. A probabilistic notion of unlearning was defined by Ginart et al. [2019], which in turn is inspired from notions in differential privacy [Dwork et al., 2006]. Several works studied algorithms for empirical risk minimization (i.e., training loss) [Guo et al., 2020, Izzo et al., 2021, Neel et al., 2021, Ullah et al., 2021], while later works study the effect of machine unlearning on the generalization loss [Gupta et al., 2021, Sekhari et al., 2021]. In particular, these works realize that unlearning data points quickly can lead to a drop in test loss, which is the theme of our current work. Several works have considered implementations of machine unlearning in several contexts starting with the work of Bourtoule et al. [2021]. These include unlearning in deep neural networks [Golatkar et al., 2020, 2021, Nguyen et al., 2020], random forests [Brophy and Lowd, 2021], large scale language models [Zanella-Béguelin et al., 2020], the tension between unlearning and privacy [Chen et al., 2021], anomaly detection [Du et al., 2019], and even auditing of machine unlearning systems [Sommer et al., 2020].

# B Poison and camouflage generation

## B.1 Gradient matching for efficient poison generation [Geiping et al., 2021]

In this section, we discuss the key intuition of Geiping et al. [2021] for efficient poison generation. Our objective is to find perturbations $\Delta$ such that when the model is trained on the poisoned samples, it minimizes the adversarial loss in (1) thus making the victim model predict the wrong label $y_{\text{adversarial}}$ on the target sample. However, directly solving (1) is computationally intractable due to bilevel nature of the optimization objective. Instead, one may implicitly minimize the adversarial loss by finding a $\Delta$ such that for any model parameter $\theta$,

$$\nabla_\theta(\ell(f(x_{\text{target}}, \theta), y_{\text{adversarial}})) \approx \frac{1}{P} \sum_{i=1}^{P} \nabla_\theta \ell(f(x^i + \Delta^i, \theta), y^i). \tag{3}$$

In essence, (3) implies that gradient based minimization (e.g., using Adam / SGD) of the training loss on poisoned samples also minimizes the adversarial loss. Thus, training a model on $S_{\text{cl}} + S_{\text{po}}$ will automatically ensure that the model predicts $y_{\text{adversarial}}$ on the target sample. Unfortunately, computing $\Delta$ that satisfies (3) is also intractable as it is required to hold for all values of $\theta$. The key idea of Geiping et al. [2021] to make poison generation efficient is to relax (3) to only be satisfied

for a fixed model $\theta_{\mathrm{cl}}$–the model obtained by training on the clean dataset $S_{\mathrm{cl}}$. To implement this, Geiping et al. [2021] minimize the cosine-similarity loss between the two gradients defined as:

$$\phi(\Delta, \theta) = 1 - \frac{\left\langle \nabla_\theta \ell(f(x_{\mathrm{target}}, \theta), y_{\mathrm{adversarial}}), \sum_{i=1}^{P} \nabla_\theta \ell(f(x_i + \Delta_i, \theta), y_i) \right\rangle}{\|\nabla_\theta \ell(f(x_{\mathrm{target}}, \theta), y_{\mathrm{adversarial}})\| \|\sum_{i=1}^{P} \nabla_\theta \ell(f(x_i + \Delta_i, \theta), y_i)\|}, \tag{4}$$

Geiping et al. [2021] demonstrated that (4) can be efficiently optimized for many popular large-scale machine learning models and datasets. For completeness, we provide their pseudocode in Algorithm 1.

## B.2 Camouflaging poisoned points

Camouflage images are designed in order to neutralize the effect of the poison images. In this section, we give intuition into what we mean by cancelling the effect of poisons, and provide two procedures for generating camouflages efficiently: label flipping, and gradient matching.

### B.2.1 Camouflages via label flipping

Suppose that the underlying task is a binary classification problem with the labels $y \in \{-1, 1\}$, and that the model is trained using linear loss $\ell(f(x, \theta), y) = -y f(x, \theta)$. Then, simply flipping the labels allows one to generate a camouflage set for any given poison set $S_{\mathrm{po}}$. In particular, $S_{\mathrm{ca}}$ is constructed as: for every $(x^i, y^i) \in S_{\mathrm{po}}$, simply add $(x^i, -y^i)$ to $S_{\mathrm{ca}}$ (i.e., $b_p = b_c$). It is easy to see that for such camouflage points, we have for any $\theta$,

$$\sum_{(x,y) \in S_{\mathrm{cpc}}} \ell(f(x, \theta), y) = -\sum_{(x,y) \in S_{\mathrm{cl}}} y f(x, \theta) - \sum_{i=1}^{P} \left( y^i f(x^i, \theta) + (-y^i) f(x^i, \theta) \right) = \sum_{(x,y) \in S_{\mathrm{cl}}} \ell(f(x, \theta), y).$$

We can also similarly show that the gradients (as well as higher order derivatives) are equal, i.e., $\nabla_\theta \sum_{S_{\mathrm{cpc}}} \ell(f(x, \theta), y) = \nabla_\theta \sum_{S_{\mathrm{cl}}} \ell(f(x, \theta), y)$ for all $\theta$. Thus, training a model on $S_{\mathrm{cpc}}$ is equivalent to training it on $S_{\mathrm{cl}}$. In essence, the camouflages have perfectly canceled out the effect of the poisons. We validate the efficacy of this approach via experiments on linear SVM trained with hinge loss (which resembles linear loss when the domain is bounded) on a binary classification problem constructed using CIFAR-10 dataset. We report the results in Table 1 (see Section **??** for details).

While label flipping is a simple and effective procedure to generate camouflages, it is fairly restrictive. Firstly, label flipping only works for binary classification problems trained with linear loss. Secondly, the attack is not clean label as the camouflage images are generated as $(x^i, -y^i)$ by giving them the opposite label to the ground truth, which can be easily caught by a validator. Lastly, the attack is vulnerable to simple data purification techniques by the victim, e.g., the victim can protect themselves by preprocessing the data to remove all the images that have both the labels ($y = +1$ and $y = -1$) in the training dataset. In the next section, we provide a different procedure to generate clean-label camouflages for general losses and multi-class classification problems.

### B.2.2 Gradient matching for generating camouflages

We next discuss our main procedure to generate camouflages, which is based on the gradient matching idea of Geiping et al. [2021]. Note that, our objective in (2) is to find $\Delta$ such that when the model is trained with the camouflages, it minimizes the original-target loss in (2) (with respect to the original label $y_{\mathrm{target}}$) thus making the victim model predict the correct label on this target sample. Since, (1) is computationally intractable, one may instead try to implicitly minimize the original-target loss by finding a $\Delta$ such that for any model parameter $\theta$,

$$\nabla_\theta (\ell(f(x_{\mathrm{target}}, \theta), y_{\mathrm{target}})) \approx \frac{1}{C} \sum_{i=1}^{C} \nabla_\theta \ell \left( f(x^i + \Delta^i, \theta), y^i \right). \tag{5}$$

(5) suggests that minimizing (e.g., using Adam / SGD) on camouflage samples will also minimize the original-target loss, and thus automatically ensure that the model predicts the correct label on the target sample. Unfortunately, (5) is also intractable as it requires the condition to hold for all $\theta$. Building on the work of Geiping et al. [2021], we relax this condition to satisfied only for a fixed

model $\theta_{\text{cp}}$-the model trained on the dataset $S_{\text{cp}} = S_{\text{cl}} + S_{\text{po}}$. Similar to what we did for generating poison points, we achieve this by minimizing the cosine-similarity loss given by

$$\psi(\Delta, \theta) = 1 - \frac{\left\langle \nabla_\theta \ell(f(x_{\text{target}}, \theta), y_{\text{target}}), \sum_{i=1}^C \nabla_\theta \ell(f(x_i + \Delta_i, \theta), y_i) \right\rangle}{\|\nabla_\theta \ell(f(x_{\text{target}}, \theta), y_{\text{target}})\| \|\sum_{i=1}^C \nabla_\theta \ell(f(x_i + \Delta_i, \theta), y_i)\|}. \tag{6}$$

**Implementation details.** We minimize (6) using the Adam optimizer [Kingma and Ba, 2015] with a fixed step size of $0.1$. In order to increase the robustness of camouflage generation, we do $R$ restarts (where $R \le 10$). In each restart, we first initialize $\Delta$ randomly such that $\|\Delta\|_\infty \le \varepsilon$ and perform $M$ steps of Adam optimization to minimize $\psi(\Delta, \theta_{\text{cp}})$. Each optimization step only requires a single differentiation of the objective $\psi$ with respect to $\Delta$, and can be implemented efficiently. After each step, we project back the updated $\Delta$ into the constraint set $\Gamma$ so as to maintain the property that $\|\Delta\|_\infty \le \varepsilon$. After doing $R$ restarts, we choose the best round by finding $\Delta_\star$ with the minimum $\psi(\Delta_\star, \theta_{\text{cp}})$. We provide the pseudocode in Algorithm 2.

---

**Algorithm 1** Gradient Matching to generate poisons [Geiping et al., 2021]

**Require:** Clean network $f(\cdot; \theta_{\text{clean}})$ trained on uncorrupted base images $S_{\text{cl}}$, a target $(x_{\text{target}}, y_{\text{target}})$ and an adversarial label $y_{\text{adversarial}}$, Poison budget $P$, perturbation bound $\varepsilon$, number of restarts $R$, optimization steps $M$

1: Collect a dataset $S_{\text{po}} = \left\{ x^i, y^i \right\}_{i=1}^P$ of $P$ many images whose true label is $y_{\text{adversarial}}$.
2: **for** $r = 1, \ldots R$ restarts **do**
3:     Randomly initialize perturbations $\Delta$ s.t. $\|\Delta\|_\infty \le \varepsilon$.
4:     **for** $k = 1, \ldots, M$ optimization steps **do**
5:         Compute the loss $\phi(\Delta, \theta_{\text{clean}})$ as in (4) using the base poison images in $S_{\text{po}}$.
6:         Update $\Delta$ using an Adam update to minimize $\phi$, and project onto the constraint set $\Gamma$.
7:     **end for**
8:     Amongst the $R$ restarts, choose the $\Delta_\star$ with the smallest value of $\phi(\Delta_\star, \theta_{\text{clean}})$.
9: **end for**
10: Return the poisoned set $S_{\text{po}} = \left\{ x^i + \Delta_\star^i, y^i \right\}_{i=1}^P$.

---

**Algorithm 2** Gradient Matching to generate camouflages

**Require:** Network $f(\cdot; \theta_{\text{cp}})$ trained on $S_{\text{cl}} + S_{\text{po}}$, the target $(x_{\text{target}}, y_{\text{target}})$, Camouflage budget $C$, perturbation bound $\varepsilon$, number of restarts $R$, optimization steps $M$

1: Collect a dataset $S_{\text{ca}} = \left\{ x^j, y^j \right\}_{j=1}^C$ of $C$ many images whose true label is $y_{\text{target}}$.
2: **for** $r = 1, \ldots R$ restarts **do**
3:     Randomly initialize perturbations $\Delta$ s.t. $\|\Delta\|_\infty \le \varepsilon$.
4:     **for** $k = 1, \ldots, M$ optimization steps **do**
5:         Compute the loss $\psi(\Delta, \theta_{\text{cp}})$ as in (4) using the base camouflage images in $S_{\text{ca}}$.
6:         Update $\Delta$ using an Adam update to minimize $\psi$, and project onto the constraint set $\Gamma$.
7:     **end for**
8:     Amongst the $R$ restarts, choose the $\Delta_\star$ with the smallest value of $\psi(\Delta_\star, \theta_{\text{cp}})$.
9: **end for**
10: Return the poisoned set $S_{\text{ca}} = \left\{ x^j + \Delta_\star^j, y^j \right\}_{j=1}^C$.

---

# C Experiment details

## C.1 Hardware

All our experiments were executed on Google Colab with a Google Colab Pro+ subscription.

## C.2 Experimental Setup

For the ease of replication, we report the corresponding poison class, target class, camouflage class and Target ID for various seeds in different experiments.

| Random Seed | Target Class | Poison Class | Camouflage Class | Target ID |
|---|---|---|---|---|
| 2000000000 | Deer | Bird | Deer | 9621 |
| 2000000001 | Cat | Horse | Cat | 1209 |
| 2000000011 | Frog | Bird | Frog | 6503 |
| 2000000111 | Bird | Cat | Bird | 124 |
| 2000001111 | Plane | Deer | Plane | 7649 |
| 2000011111 | Cat | Dog | Cat | 4423 |
| 2000111111 | Truck | Car | Truck | 8117 |
| 2001111111 | Bird | Truck | Bird | 3686 |
| 2011111111 | Cat | Bird | Cat | 642 |
| 2111111111 | Frog | Ship | Frog | 97 |

Table 2: Target, poison and camouflage class corresponding to different initial random seeds used for CIFAR-10 experiments. The reported Target ID is relative to the CIFAR-10 validation dataset.

| Random Seed | Target Class | Poison Class | Camouflage Class | Target ID |
|---|---|---|---|---|
| 2000000000 | Building | Cassette player | Building | 1559 |
| 2000000001 | Chain saw | Gas pump | Chain saw | 1266 |
| 2000000011 | Truck | Cassette player | Truck | 2460 |
| 2000000111 | Cassette player | Chain saw | Cassette player | 792 |
| 2000001111 | Tench | Building | Tench | 2500 |
| 2000011111 | Chain saw | French horn | Chain saw | 1162 |
| 2000111111 | Parachute | English springer | Parachute | 3826 |
| 2001111111 | Cassette player | Parachute | Cassette player | 1121 |
| 2011111111 | Chain saw | Cassette player | Chain saw | 1198 |
| 2111111111 | Truck | Golf ball | Truck | 2343 |

Table 3: Target class, poison class and camouflage class corresponding to different random seeds used for Imagenette experiments. The reported target ID is relative to the Imagenette validation set.

| Random Seed | Target Class | Poison Class | Camouflage Class | Target ID |
|---|---|---|---|---|
| 2000000000 | Border Terrier | Beagle | Border Terrier | 1493 |
| 2000000001 | English Foxhound | Old English Sheep Dog | English Foxhound | 1362 |
| 2000000011 | Golden Retriever | Beagle | Golden Retriever | 2399 |
| 2000000111 | Beagle | English Foxhound | Beagle | 827 |
| 2000001111 | Shih-Tzu | Border Terrier | Shih-Tzu | 250 |
| 2000011111 | English Foxhound | Australian Terrier | English Foxhound | 1405 |
| 2000111111 | Dingo | Rodesian Ridgeback | Dingo | 3810 |
| 2001111111 | Beagle | Dingo | Beagle | 1204 |
| 2011111111 | English Foxhound | Beagle | English Foxhound | 1294 |
| 2111111111 | Golden Retriever | Samoeyed | Golden Retriever | 2282 |

Table 4: Target class, poison class and camouflage class corresponding to different random seeds used for Imagewoof experiments. The reported target ID is relative to the Imagewoof validation set.

# D   Main Experiments

In this section, we give details into our experimental setup. We generate poison points by running Algorithm 1, and camouflage points by running Algorithm 2 with $R = 1$ and $M = 250$.[5] Each experiment is repeated $K$ times by setting a different seed each time, which fixes the target image, poison class, camouflage class, base poison images and base camouflage images. Due to limited computation resources, we typically set $K \in \{3, 5, 8, 10\}$ depending on the dataset and report the

---

[5]We note that we diverge slightly from the threat model described above, in that the adversary *modifies* rather than introduces new points. We do this for convenience, but we do not anticipate the results would qualitatively change.

mean and standard deviation across different trials. We say that *poisoning* was successful if the model trained on $S_{\text{cp}} = S_{\text{cl}} + S_{\text{po}}$ predicts the label $y_{\text{adversarial}}$ on the target image. Furthermore, we say that *camouflaging* was successful if the model trained on $S_{\text{cpc}} = S_{\text{po}} + S_{\text{cl}} + S_{\text{ca}}$ predicts back the correct label $y_{\text{target}}$ on the target image, provided that poisoning was successful. A camouflaged poisoning attack is successful if both poisoning and camouflaging were successful.

## D.1 Evaluations on Cifar-10

We extensively evaluate our camouflaged poisoning attack on models trained on the CIFAR-10 dataset [Krizhevsky, 2009]. CIFAR-10 is a multiclass classification problem with 10 classes, with 6,000 color images in each class (5,000 training + 1,000 test) of size $32 \times 32$. We follow the standard split into 50,000 training images and 10,000) validation / test images.

### D.1.1 Support Vector Machines

In order to perform evaluations on SVM, we first convert the CIFAR-10 dataset into a binary classification dataset (which we term as Binary-CIFAR-10) by merging the 10 classes into two groups: `animal` ($y = +1$) and `machine` ($y = -1$). Images (in the training and the test dataset) that were originally labeled (*bird, cat, deer, dog, frog, horse*) are instead labeled `animal`, and the remaining images, with original labels (*airplane, cars, ship, truck*), are labeled `machine`.

We train a linear SVM (no kernel was used) with the hinge loss: $\ell(f(x, \theta), y) = \max\{0, 1 - yf(x, \theta)\}$. The training was done using the `svm.LinearSVC` class from Scikit-learn [Pedregosa et al., 2011] on a single CPU. In the pre-processing stage, each image in the training dataset was normalized to have $\ell_2$-norm 1. Each training on Binary-CIFAR-10 dataset took 25 - 30 seconds. In order to generate the poison points, we first use `torch.autograd` to compute the cosine-similarity loss (4), and then optimize it using Adam optimizer with learning rate 0.001. Each poison and camouflage generation took about 40 - 50 seconds (for $b_p = b_c = 0.2\%$). We evaluate both label flipping and gradient matching to generate camouflages, and different threat models ($\varepsilon, b_p, b_c$); the results are reported in Table 1. For each of our experiments we chose $K = 10$ seeds. Each trained model had validation accuracy of around 81.63% on the clean dataset $S_{\text{cl}}$, which did not change significantly when we retrained after adding poison samples and / or camouflage samples. Note that the efficacy of the camouflaged poisoning attack was more than 70% in most of the experiments. We provide a sample of the generated poisons and camouflages in Figure 6.

### D.1.2 Neural Networks

We perform extensive evaluations on the multiclass CIFAR-10 classification task with various popular large scale neural networks architectures including VGG-11, VGG-16 [Simonyan and Zisserman, 2015], ResNet-18, ResNet-34, ResNet-50 [He et al., 2016], and MobileNetV2 [Sandler et al., 2018].

Each model is trained with cross-entropy loss $\ell(f(x, \theta), y) = -\log(\Pr(y = f(x, \theta)))$ on a single GPU using PyTorch [Paszke et al., 2019], and using mini-batch SGD with weight decay 5e-4, momentum 0.9, learning rate 0.01, batch size 100, and 40 epochs over the training dataset. Each training run took about 45 minutes. The poison and camouflage sets were generated using gradient matching by first defining the cosine-similarity loss using `torch.autograd` and then minimizing it using Adam with a learning rate of 0.1. Each poison/camouflage generation took about 1.5 hours.

We report the efficacy of our camouflaged poisoning attack across different models and threat models ($\varepsilon, b_p, b_c$) in Figure 3; also see Appendix D.3 for detailed results and performance drops on the validation dataset after adding poison and camouflage set. Each model was trained to have validation accuracy between 81-87% (depending on the architecture), which changed minimally when the model was retrained with poison and camouflage samples. Poisoning was successful at least 80% of the time in most of the experiments. Camouflaging was successful at least 70% of the time for VGG-11, VGG-16, Resnet-18, and Resnet-34 but was not as successful for MobileNetV2 and Resnet-50. Furthermore, camouflaging succeeded at least 75% of times when $b_c = b_p$, but did not perform as well when we set $b_p > b_c$ in the thread model (more poison images than camouflage images).

## D.2 Evaluations on Imagenette and Imagewoof

We evaluate the efficacy of our attack vector on the challenging multiclass classification problem on the Imagenette and Imagewoof datasets [Howard, 2019]. Imagenette is a subset of 10 classes (*Tench, English springer, Cassette player, Chain saw, Building/church, French horn, Truck, Gas pump, Golf ball, Parachute*) from the Imagenet dataset [Russakovsky et al., 2015]. The Imagenette dataset consists of around 900 images of various sizes for each class. In total, we have 13394 images which are divided into a training dataset of size 9469 and test dataset of size 3925. To perform training, all images are resized and centrally cropped down to $224 \times 224$ pixels.

| Dataset | Model | Threat Model | | | Attack Success | |
|---|---|---|---|---|---|---|
| | | $\varepsilon$ | $b_p$ | $b_c$ | Poisoning | Camouflaging |
| Imagenette | VGG-16 | 16 | 6.3% | 6.3% | 25% | 100% |
| Imagenette | Resnet-18 | 16 | 6.3% | 6.3% | 40% | 50% |
| Imagewoof | Resnet-18 | 16 | 6.6% | 6.6% | 50% | 75% |

Table 5: Evaluation of camouflaged poisoning attack on Imagenette and Imagewoof datasets over 5 seeds (with 1 restart per seed). Note that camouflaging succeeded in most of the experiments in which poisoning succeeded. Prior works (e.g., Geiping et al. [2021]) set a large number of restarts $R$, and then choose the most effective attack among them. Due to computational constraints, we ran only one restart (i.e., $R = 1$) for each experiment. Given additional computational resources, we could inflate the success rate of both the poisoning and camouflaging.

Imagewoof [Howard, 2019] is another subset of Imagenet dataset consisting of 10 classes (*Shih-Tzu, Rodesian Ridgeback, Beagle, English Foxhound, Border Terrier, Austrailian Terrier, Golden Retriever, Old English Sheep Dog, Samoyed, Dingo*). Imagewoof consists of around 900 images of various sizes for each class, and in total 12954 images which are divided into a training dataset of size 9025 and test dataset of size 3929. Similar to Imagenette, we resize all images and crop to the central $224 \times 224$ pixels before training.

We evaluate our camouflaged poisoning attack on two different neural network architectures-VGG-16 and ResNet-18, and different threat models $(\varepsilon, b_p, b_c)$ listed in Table D.2. Each model is trained on a single GPU with cross-entropy loss, that is minimized using SGD algorithm with weight decay 5e-4, momentum 0.9 and batch size 20. We start with a learning rate of 0.01, and exponentially decay it with $\gamma = 0.9$ after every epoch, for a total of 50 epochs over the training dataset. The poisons and camouflages were generated using gradient matching by first defining the cosine-similarity loss using `torch.autograd` and then optimizing it using Adam optimizer with learning rate 0.1. In our experiments, camouflaging was successful for at least $50\%$ of the time when poisoning was successful. However, because we modified about 13% of the training dataset when adding poisons / camouflages, we observe that the fluctuation in the model's validation accuracy can be up to 7% for both Imagenette and Imagewoof, as expected on making such a large change in the training set.

## D.3 Additional details on CIFAR-10 Experiments on neural networks

We elaborate on the results reported in Figure 3. In Table 6, we report the efficacy of the proposed camouflaged poisoning attack on different neural network architectures where the threat model is given by $\varepsilon = 16, b_p = 0.6\%, b_c = 0.6\%$. The reported results are an average over 5 seeds from 2000000000-2000001111. In the first column under attack success, we report the number of times poisoning was successful amongst the run trials, and in the second column, we report the number of times camouflaging was successful for the trials for which poisoning was successful.

In Table 7, we report the success of the proposed attack when we change the threat model, but fix the network architecture to be ResNet-18. Each experiment was repeated times 5 times with 8 restarts each time, and the mean success rate is reported. These experiments were conducted with 5 seeds from 2000011111-2111111111.

| Network Architecture | Attack success | | Validation Accuracy | | |
| --- | --- | --- | --- | --- | --- |
| | Poisoning | Camouflaging | Clean | Poisoned | Camouflaged |
| VGG-11 | 100% | 80% | 85.01 | 85.03 (± 0.37) | 85.10 (± 0.29) |
| VGG-16 | 80% | 75% | 87.68 | 87.42 (± 0.17) | 87.45 (± 0.26) |
| ResNet-18 | 80% | 75% | 82.13 | 81.88 (± 0.15) | 81.80 (± 0.12) |
| ResNet-34 | 80% | 50% | 82.45 | 82.61 (± 0.30) | 83.12 (± 0.93) |
| ResNet-50 | 80% | 25% | 81.02 | 81.76 (± 0.13) | 84.62 (± 0.71) |
| MobileNetV2 | 60% | 33% | 82.79 | 83.26 (± 0.25) | 85.47 (± 0.27) |

Table 6: Evaluating our proposed camouflaged poisoning attack on various model architectures on the CIFAR-10 dataset with the threat model $\varepsilon = 16, b_p = 0.6\%, b_c = 0.6\%$.

| Threat model | | | Attack success | | Validation Accuracy | | |
| --- | --- | --- | --- | --- | --- | --- | --- |
| $\varepsilon$ | $b_p$ | $b_c$ | Poisoning | Camouflaging | Clean | Poisoned | Camouflaged |
| 16 | 1% | 1% | 100% | 80% | 82.13 | 81.98 (± 0.16) | 82.12 (± 0.21) |
| 8 | 1% | 1% | 80% | 75% | 82.13 | 82.21 (± 0.21) | 82.09 (± 0.23) |
| 16 | 2% | 1% | 100% | 20% | 82.13 | 82.31 (± 0.26) | 82.19 (± 0.24) |
| 8 | 2% | 1% | 100% | 40% | 82.13 | 82.43 (± 0.30) | 82.34 (± 0.27) |

Table 7: Evaluating our proposed camouflaged poisoning attack on various threat models with CIFAR-10 dataset trained on ResNet-18.

## D.4 Visualizations

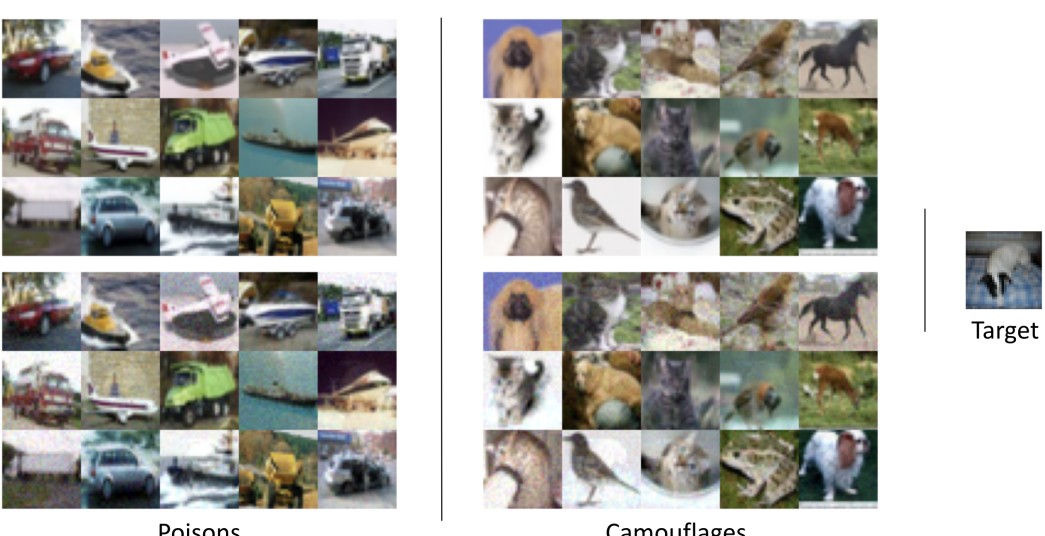

Poisons          Camouflages          Target

Figure 6: Visualization of poisons and camouflages on Binary-CIFAR-10 dataset (*animal* vs *machine* classification). The top row shows the original images and the bottom row shows the corresponding poisoned / camouflaged images (with the added $\Delta$). The shown images were generated for a camouflaged poisoning attack on SVM, with Seed = 555555, $\varepsilon = 16, b_p = 0.2, b_c = 0.4$ and the target ID 6646.

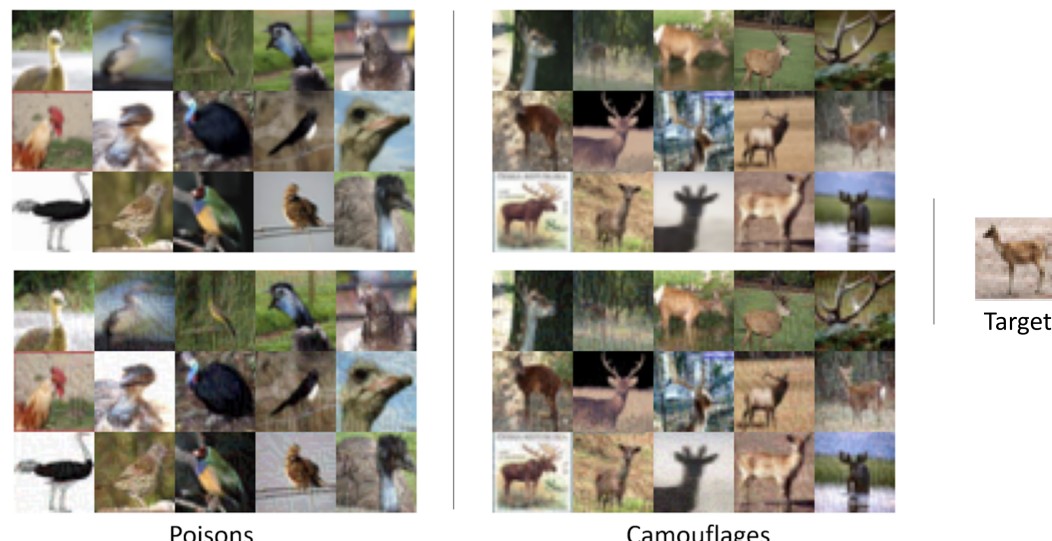

Poisons          Camouflages          Target

Figure 7: Visualization of poisons and camouflages on CIFAR-10 dataset (multiclass classification task). The top row shows the original images and the bottom row shows the corresponding poisoned / camouflaged images (with the added $\Delta$). The shown images were generated for a camouflaged poisoning attack on ResNet-18, with Seed = 2000000000, $\varepsilon = 8$, $b_p = 0.2$, $b_c = 0.4$, poison class *bird*, target class *deer*, and the target ID 9621.

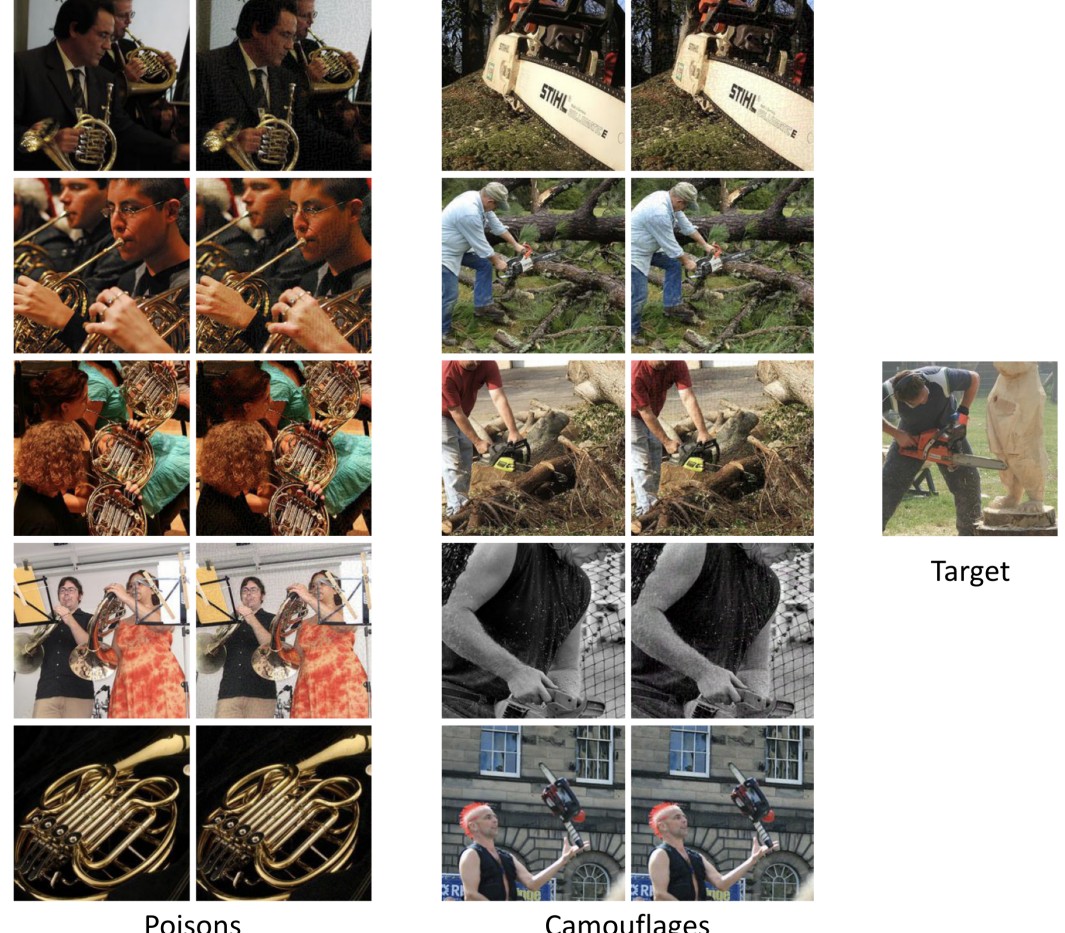

Target

Poisons                    Camouflages

Figure 8: Visualization of poisons and camouflages on Imagenette dataset. The first and the third columns shows the original images, and the second and the fourth columns shows the corrupted images (with added $\Delta$). The shown images were generated for a camouflaged poisoning attack on ResNet-18, with Seed = 2000011111 and $\varepsilon = 8$. The target and camouflage class is *chain saw*, and the poison class is *French horn*.

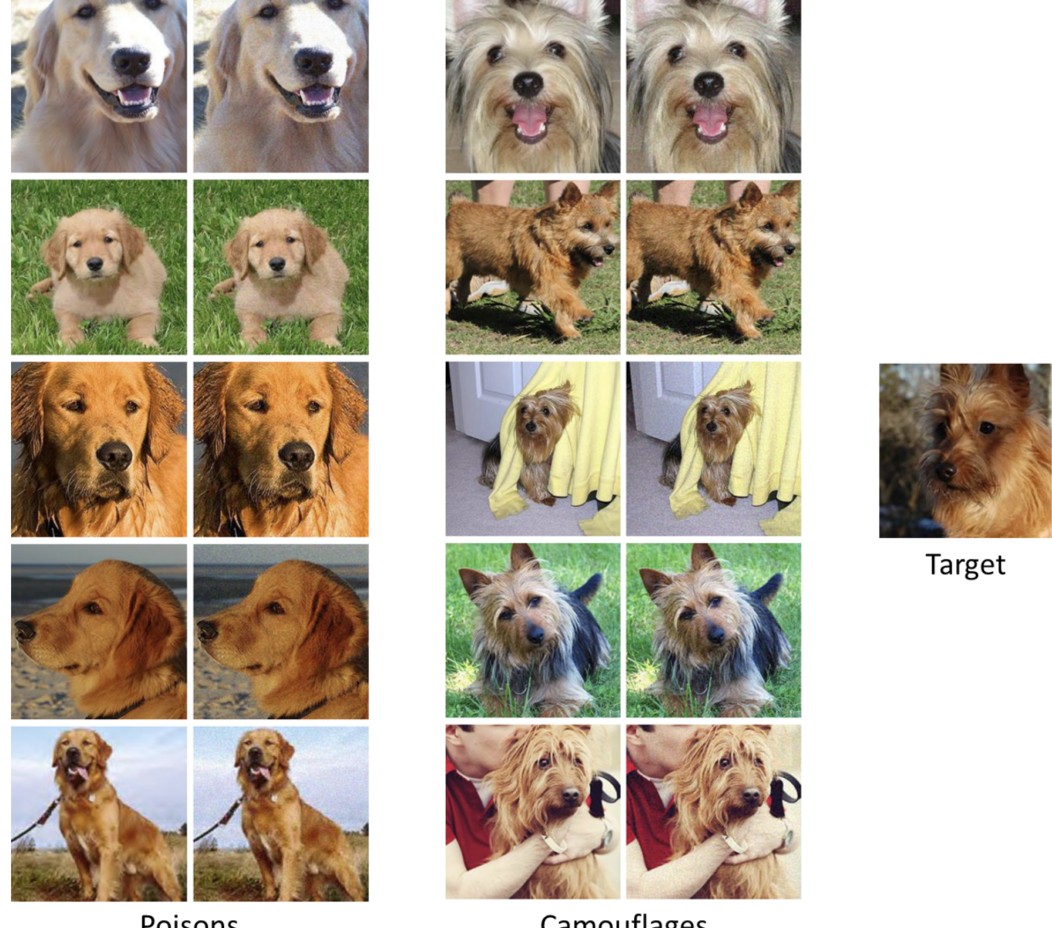

Poisons

Camouflages

Target

Figure 9: Visualization of poisons and camouflages on Imagewoof dataset. The first and the third columns shows the original images, and the second and the fourth columns shows the corrupted images (with added $\Delta$). The shown images were generated for a camouflaged poisoning attack on ResNet-18, with Seed = 2111111110, $b_p = b_c = 4.2\%$, $\varepsilon = 16$. The target and camouflage class is *Austrailian Terrier*, and the poison class is *Golden Retriever*.

### D.5 Further Experiments

In Appendix D.5, we provide additional experiments on CIFAR-10 showing that our attack is robust to data augmentation, and successfully transfers when the victim model is different from the model on which poison and camouflage samples were generated.

#### D.5.1 Transfer experiments

In this section, we show that the poison and camouflage samples generated by the proposed approach transfer across models. Thus, an attacker can successfully execute the camouflaged poisoning attack, even if the victim trains a different model than the one on which the poison and camouflage samples were generated. We show the transfer success in Figure 10. The brewing network denotes the network architecture on which poison and camouflage samples were generated (we adopt the same notation as Geiping et al. [2021]). The victim network denotes the model architecture used by the victim for training on the manipulated dataset.

We ran a total of 3 experiments per (brewing model, victim model) pair using the seeds 2000000000-2000000011. Each reported number denotes the fraction of times when both poisoning and camouflaging were successful in the transfer experiment, and thus the attack could take place.

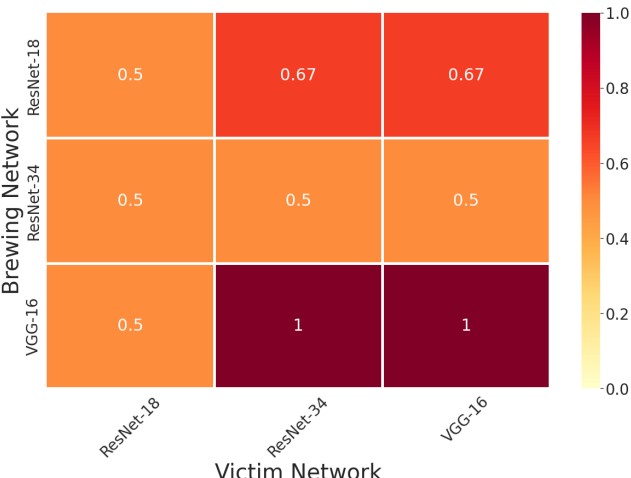

Figure 10: Transfer experiments on CIFAR-10 dataset.

#### D.5.2 Robustness to Data Augmentation

Data augmentation is commonly used to avoid overfitting in deep neural networks. In order to be applicable in the real life, our poisoning and camouflaging attacks must be successful even when the model is trained with data augmentation. In order to validate this, we evaluate our approach on CIFAR-10 dataset trained with data augmentation on ResNet-18 in the threat model $\varepsilon = 16, b_p = b_c = 1\%$; the results are in Table 8. The considered data augmentations are:

1. *No Augmentation*: Exact images from the training dataset are used.

2. *Augmentation Set 1*: $50\%$ chance that the image will be horizontally flipped, but no rotations.

3. *Augmentation Set 2*: $50\%$ chance that the image will be horizontally flipped, and random rotations in Uniform$(-10, 10)$ degrees.

The reported results in Table 8 are an average over 5 random seeds from "kkkkkk" where $1 \le k \le 5$. As expected, the validation accuracy for the model trained on clean dataset increased from 82% percent when trained without augmentation, to 86% for augmentation Set 1 and 88% for augmentation set 2. The addition of data augmentation during training and re-training stages make it harder for poisoning to succeed and at the same time makes it easier for camouflaging to succeed.

| Data Augmentation | Attack success | | Validation Accuracy | | |
|---|---|---|---|---|---|
| | Poisoning | Camouflaging | Clean | Poisoned | Camouflaged |
| No Augmentation | 100% | 20% | 82% | 82% | 82% |
| Augmentation Set 1 | 86% | 33% | 86% | 85% | 86% |
| Augmentation Set 2 | 60% | 100% | 88% | 86 | 86% |

Table 8: Effect of data augmentation on our proposed camouflaged poisoning attack.

### D.5.3 Similarity of the feature space distance

A natural approach to defend against dataset manipulation attacks is to try to identify the modified images, and then remove them from the training dataset (i.e., *data sanitization*). For instance, one could cluster images based on their distance from their class mean image, or from the target image. This type of defense could potentially thwart watermarking poisoning attacks such as Poison Frogs [Shafahi et al., 2018]. As we show in Figure 11, such a defense would not be effective against our proposed poison and camouflage generation procedures, as the data distribution for the poison set and the camouflage set is similar to that of the clean images from the respective classes.

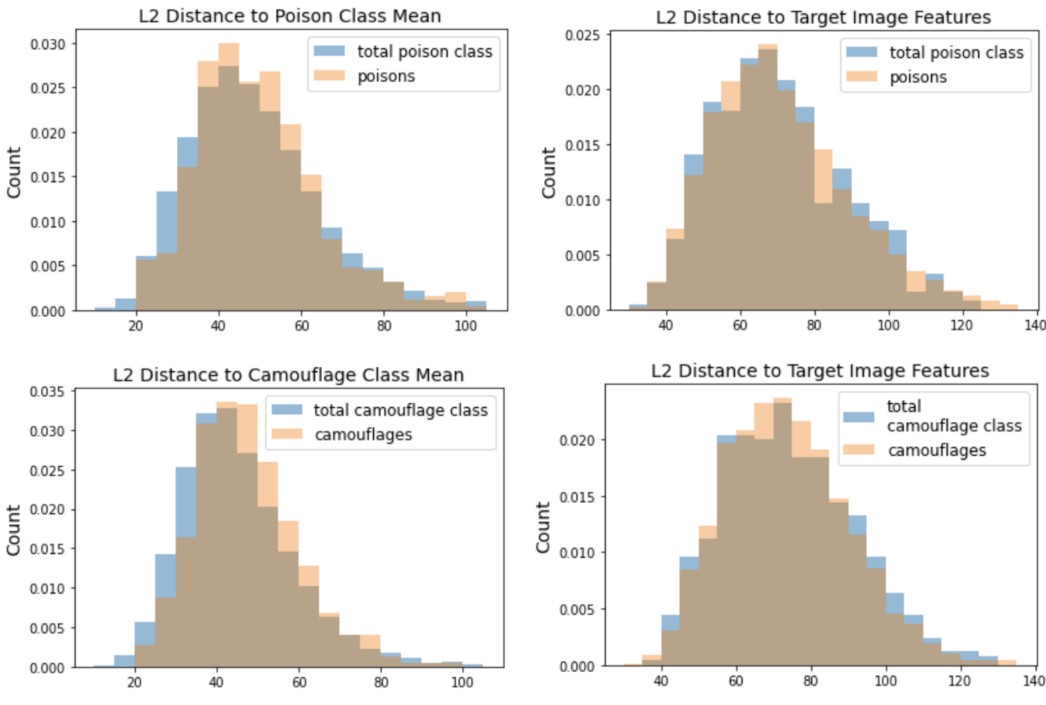

Figure 11: Feature space distance for our generated poison and camouflage set. The reported data was collected by a successful camouflaged poisoning attack on Resnet-18 model trailed on CIFAR-10 with seed 2000000000, $\varepsilon = 16$ and $b_p = b_c = 1\%$.

