# OpenReview forum: "Hidden Poison: Machine Unlearning Enables Camouflaged Poisoning Attacks"
_NeurIPS.cc/2022/Workshop/TSRML — TSRML2022_

### Official Review · Reviewer_hHLV · 2022-10-17
**A good paper proposing a novel attack named camouflaged poisoning**

**Overall Rating:** 7

**Summary:**

The paper proposes a novel poisoning attack named camouflaged poisoning. It builds upon previous data poisoning literature and the key idea is to add a small adversarial perturbation in addition to the poisons to a subset of a dataset (called camouflage samples) so that the attack is activated when the camouflage samples are removed from the trainset and the network is trained on this modified trainset.

**Strengths:**

- The paper is well-written and easy to follow.
- The proposed camouflaged poisoning attack is novel and interesting.
- The camouflaged poisoning could be potentially more dangerous and more practical than conventional poisoning attacks.
- The attack is evaluated on various network architectures and datasets.

**Weaknesses:**

- Even though the idea of camouflaged poisoning attack is novel, the poisoning technique heavily relies on Witches' Brew which is not novel.
- In practical settings, the attacker typically perturbs $\le1$% of the trainset. However, the proposed attack requires two subsets of perturbations for poisons and camouflage samples separately, this may be a drawback.
- The success rate is relatively low for some network architectures (e.g. ResNet-34, ResNet-50 and MobileNetV2)
- The paper's organization should be enhanced. I believe the authors could replace Figures 5 and 6 with Main Experiments (Section D) in the main body of the paper.


--
Please update the manuscript with the following modifications:

- Please cite the recent relevant papers (e.g., [1-4]).
- There are no references to Table 1, Figures 2 and 3 in the main body of the paper. Please add some explanations.
- I assume that, similar to previous work, perturbations are added to a subset of the trainset and replaced with clean base images. Therefore the size of trainset remain unchanged. However, from the notations $S cpc = S cl + S po + S ca$ and $S cp = S cl + S po$, it seems that the poisons and camouflage samples are extra samples added to the trainset.  Please make sure to clarify this.
- When removing camouflage samples, are only the perturbations removed (while keeping the clean base samples) or the samples are removed entirely from the trainset? Please explain further.
- Please include evaluations of $b p = b c = 0.5 \\%$. This is usually the standard budget.
- Please mention the number of restarts for the experiments depicted in Figure 3 and Table 6.
- Please provide the number of trials and the standard error for the main experiments.
- In Algorithms 1 and 2, the loss functions are not referenced correctly.
- A reference is missing in section B.2.1.
- Typo in: "Each experiment was repeated times 5 times with 8 restarts each time ..."


1- Fowl, Liam, Micah Goldblum, Ping-yeh Chiang, Jonas Geiping, Wojtek Czaja, and Tom Goldstein. "Adversarial examples make strong poisons."

2- Souri, Hossein, Micah Goldblum, Liam Fowl, Rama Chellappa, and Tom Goldstein. "Sleeper agent: Scalable hidden trigger backdoors for neural networks trained from scratch."

3- Zhu, Chen, W. Ronny Huang, Hengduo Li, Gavin Taylor, Christoph Studer, and Tom Goldstein. "Transferable clean-label poisoning attacks on deep neural nets."

4- Guo, Junfeng, and Cong Liu. "Practical poisoning attacks on neural networks."





**Overall Recommendation:**

This paper proposes a novel poisoning attack named camouflaged poisoning. The camouflaged poisoning could be potentially dangerous and practical. Also, extensive evaluations of various models and datasets were carried out by the authors. While this is generally a well-written paper, it still requires some modifications. I vote to accept this paper. Please make sure to address my concerns outlined in the Weaknesses.

**Review Confidence:**

5: The reviewer is absolutely certain that the evaluation is correct and very familiar with the relevant literature

---

### Official Review · Reviewer_JWbH · 2022-10-19
**Interesting attack, but should be better motivated**

**Overall Rating:** 6

**Summary:**

The paper proposes a "camouflage" attack, which leverages a novel threat vector introduced by machine unlearning. An attacker, with white box access to a model, adds two types of manipulated data -- poison and camouflage -- to the model's training dataset. A model trained on poison+camouflage samples should behave normally. The attacker then triggers removal of the camouflage samples via an unlearning request. When the model is retrained from scratch on *only* the poison data, the attacker can effect a targeted misclassification of a single data point.

**Strengths:**

- This paper provides thorough background and experimental evaluation of the attack.
- The unlearning-based threat vector is novel, and the authors propose an interesting way to exploit it.
- Tables/Figures are clear and well-marked.

**Weaknesses:**

There are three main weaknesses the authors resolve in future versions of this paper.

- The execution of this paper is good -- good experiments and theory, interesting result. However, I'm struggling with the *why* of this paper. *Why* would an attacker choose this approach, as opposed to just inserting the poisoning points on their own from the get-go? *When* in the real world would this attack make sense?  You mention briefly (lines 109-112) that camouflage attacks are *more dangerous* because the attacker can choose when they happen. Justify this claim. The motivation must be stronger, otherwise people will likely just see it as an incremental extension of [Geiping et al, 2021](https://arxiv.org/abs/2009.02276).

- I'm not convinced that this attack will work as well in the approximate unlearning use case vs. the training from scratch one. The authors need to justify their claim in lines 60-62 that the "phenomenon should still occur" in approximate (vs. exact) unlearning settings. Approximate unlearning on poisoned or manipulated data is still an unexplored space. Such data points may affect the model differently than normal data, resulting in an unexpected attack result. I think adding results on approximate unlearning would greatly strengthen the paper (and probably open new, interesting research questions about poisoning + unlearning in general).

- Minor points:
    - Can the authors comment on why poison/camouflage success rate decreases as model size increases (e.g. Table 6)? What does this mean for the scalability of this attack?
    - Figure 2 is never referred to in the text. What is the epsilon for the poison/camouflage images shown here?

**Overall Recommendation:**

I recommend accepting this paper, but I think for the paper to be accepted to a conference (where I assume it will be submitted in the future) I recommend that the authors address the weaknesses I raised:
1. Develop a stronger motivation for this attack. The current attack assumes a very specific scenario (e.g. unlearning requests are made/honored, full retraining as the unlearning setting) that doesn't yet (and may never) exist. It furthermore provides no justification for why the attacker would go through this multi-step attack in the first place, as opposed to just inserting the poison data a priori.
2. Evaluate in multiple unlearning scenarios, rather than claiming that performance under a particular unlearning approach (full retraining) will generalize.

**Review Confidence:**

4: The reviewer is confident but not absolutely certain that the evaluation is correct

---

### Decision · Program_Chairs · 2022-10-23

**Decision:**

Accept

**Comment:**

Following the unanimous recommendations from reviewers, the submission is accepted.